# Miniaturized Band Pass Filter Design Using Half Mode Substrate Integrated Coaxial Resonators

**DOI:** 10.3390/mi13030389

**Published:** 2022-02-28

**Authors:** Min-Hua Ho, Chung-I G. Hsu, Kun-Hua Tang, Wanchu Hong

**Affiliations:** 1Department of Electronic Engineering, National Changhua University of Education, Changhua City 50074, Taiwan; whong@cc.ncue.edu.tw; 2Department of Electrical Engineering, National Yunlin University of Science and Technology, Douliou City 64002, Taiwan; cghsu@yuntech.edu.tw; 3Advanced Ceramic X Corporation, Hsinchu City 30352, Taiwan; skt297y6@gmail.com

**Keywords:** half-mode substrate integrated coaxial resonator (HMSICR), substrate integrated coaxial resonator (SICR), substrate integrated waveguide (SIW), circuit miniaturization, band-pass filter (BPF)

## Abstract

The contribution of this work is to propose a half-mode substrate integrated coaxial resonator (HMSICR) and its application in bandpass filter (BPF) design. The proposed HMSICR is formed by evenly bisecting a square substrate integrated coaxial resonator (SICR), which is a cavity composed of two dielectric substrates and three metal layers. The SICR’s sidewalls are mimicked by periodically spaced thru-via arrays, and a circular patch is embedded in the middle metal layer of the SICR with the patch shorted to the cavity’s bottom wall by a circular array of blind vias. This HMSICR can drastically lower the cavity’s resonance frequency. The achieved frequency reduction rate of the proposed HMSICR, as compared with that of its conventional substrate integrated waveguide (SIW) cavity counterpart, reaches 70%. A sample four-HMSICR BPF is built for the circuit verification measurement. To further reduce the sample filter’s area, the composing HMSICRs are vertically stacked in a back-to-back configuration. We believe that its obtained size-reduction rate reaches the highest record.

## 1. Introduction

The resonator cavity plays a very important role in building BPFs of wireless communication systems, especially low-loss, high-Q, and compact BPFs. In the early years, most microwave resonators were composed of rectangular or cylindrical waveguide cavities. The cavity’s resonance frequency depends on the size of the cavity, and a larger cavity leads to a lower resonance frequency. Normally, waveguide cavities of various forms are the essential choices to serve as a resonator [1]. However, the waveguide’s drawbacks of bulky volume, heavy weight, and high cost have hindered waveguide cavities from being employed in commercial communication systems that require small, light, and cheap interior components. In this paper, we propose a novel cavity resonator that overcomes the waveguide’s drawbacks. The proposed cavity resonator is constructed using printed circuit board (PCB) technology, which is cost-effective in fabrication and suitable for massive production.

Two decades ago, a novel structure named the substrate integrated waveguide (SIW) [2,3,4,5] built by standard PCB technology was proposed. Although the SIW that emulates a metal waveguide is non-bulky and still possesses a reasonably high power-handling capability, the SIW resonator is still considered too large to be used for building compact microwave devices. To alleviate the large-size drawback of the SIW resonator, several pilot research works have been conducted to miniaturize SIW resonators [6,7,8,9,10]. In [6], the SIW is allowed to operate under the cut-off frequency with its broadside wall embedded with complementary split-ring resonators (CSRRs). In [7], the embedded CSRRs can reduce the fundamental-mode resonance frequency of an SIW cavity. As a result, the circuit size is reduced by the effect of frequency reduction. The size reduction can also be achieved by using the waveguide’s (or cavity’s) fractional-mode variants, e.g., the half-mode SIW [8] and quarter-mode SIW cavities [9,10]. Besides the size reduction, a wide bandwidth (BW) can also be observed in [10] owing to its triple-mode resonance. However, the radiation loss from the fractional-mode SIW’s open sides is larger at a higher operating frequency, and the circuit-size reduction is quite limited. Although the above-mentioned structures have achieved the size reduction to a certain degree, the circuit dimensions are still too large for commercial applications.

Recently, size reduction for SIWs has been persistently investigated using the evanescent-mode design (or the so-called coaxial-mode design) [11,12]. The obtained size-reduction rate in [11] is 67%, and that in [12] arrives at a record high of 96.7% for a single-unit coaxial cavity. Although the substrate integrated coaxial resonator (SICR) in [12] has a very high size-reduction rate, its step-impedance structure consisting of three dielectric layers requires a costly PCB fabrication process. Later, a similar SICR that requires a less complicated PCB process was proposed in [13], whose size-reduction rate reaches a comparable value of 96.4%. In addition, the spurious passbands of the BPFs in [13] are relatively farther away from the working passband than those of other full-mode-based SIW BPFs. In this paper, the fractional-mode technique and the evanescent-mode design are combined to form a half-mode substrate integrated coaxial resonator (HMSICR), which is then adopted to build a better size-efficient BFP. A sample four-HMSICR BPF is designed and fabricated for circuit-performance validation. To further reduce the circuit size, the four HMSICRs in the BPF are divided into two identical sets with each set composed of two side-by-side HMSICRs, and these two sets are then vertically stacked in a back-to-back form, thus cutting off an extra circuit area by half.

## 2. Half-Mode Substrate Integrated Coaxial Resonator

The proposed HMSICR is originated from the full-mode SICR given in Figure 1. This full-mode SICR is composed of three metal layers and two tightly stacked substrates with the same dielectric constant (*ε_r_*) and loss tangent (tan*δ*). The thicknesses of the top and bottom substrates are denoted by *h*_1_ and *h*_2_, respectively. The cavity’s sidewalls are emulated by four periodically spaced thru-via arrays. The key point in miniaturizing the cavity is to reduce the cavity’s resonance frequency. This is accomplished by introducing a short-circuited circular metal patch in the cavity. The embedded circular patch is deployed in the middle layer (see Figure 1) and a circular array of blind vias is used to short the patch to the cavity’s bottom wall. The function of the embedded patch is to load the cavity with a large capacitance, which helps lower the resonance frequency [14,15]. It is expected that a cavity loaded with a larger capacitance will result in a lower resonance frequency. In Figure 2, we give the simulated normalized resonance frequency (*f_r_*/*f*_0_) and unloaded Q-factor (*Q_u_*) as functions of *h*_1_. In the simulation of this figure, the substrate is RT/duroid 5880 with *ε_r_* = 2.2 and tan*δ* = 0.0009, which will also be adopted for our proposed HMSICR design. In addition, the metal sheets in the three metal layers all have the same thickness of 35 *μ*m, the square cavity has a side length of *L* = 24 mm, the circular patch has a radius of *R_p_* = 11 mm, the circular blind-via array has a radius of *R_v_* = 5.5 mm, and the bottom substrate has a thickness of *h*_2_ = 1.58 mm. Here, *f_r_* is the resonance frequency of the full-mode SICR, and *f*_0_ = 5.97 GHz is the fundamental-mode (TE_101_-mode) resonance frequency of the conventional SIW cavity counterpart (i.e., the SIW cavity devoid of the circular patch and circular array of blind vias). It is observed from Figure 2 that a smaller value of *h*_1_ (corresponding to a larger loaded capacitance) leads to a lower resonance frequency. The fact that *f_r_* is much smaller than *f*_0_ indicates that the full-mode SICR structure can effectively miniaturize the SIW cavity resonator. Although the goal in lowering the resonance frequency is achieved in this structure, the drawback is that the full-mode SICR has a smaller *Q_u_* value than does its conventional SIW cavity counterpart having a *Q_u_* value of around 680. In Figure 3, we show the simulated curves of *f_r_*/*f*_0_ as functions of *R_p_* and *R_v_*. In the simulation, *h*_1_ is fixed at 0.254 mm, and other relevant structural parameters are the same as those used in the simulation of Figure 1. A larger value of *R_p_* provides the cavity with a larger loaded capacitance and hence results in a lower resonance frequency. On the contrary, a larger value of *R_v_* leads to a higher resonance frequency. This trend is opposite to that of the *R_p_* variation and can be explained using a transmission-line model presented later for the HMSICR proposed in this section.

As shown in Figure 4, the proposed HMSICR is obtained by evenly bisecting the full-mode SICR given in Figure 1, and the layouts of the HMSICR’s three metal layers are given in Figure 5. The substrates adopted here are also RT/duroid 5880 and have the same thicknesses as those associated with Figure 3. Each of the blind vias has a diameter of 0.6 mm. The three thru-via arrays (also with a diameter of 0.6 mm for each via) mimic the three sidewalls of the cavity, and the fourth side of the cavity is open. The periodicity of the thru-via array is 2 mm. The open cavity in Figure 5 has the dimension of 12 × 24 mm^2^ (i.e., *L* = 24 mm in Figure 5).

Note that the full-mode SICR operates at a coaxial mode whose resonance-frequency expression developed from a transmission-line model can be found in [13]. Here, the HMSICR has one open side, which makes the field distribution under the semicircular patch slightly different from that of a coaxial-line mode. Hence, the expression needs to be modified to estimate the resonance frequency of the HMSICR. Figure 6 shows the side view of the HMSICR and the equivalent transmission-line model. The susceptance, *B*(*ω*), looking into the circular patch can be expressed as
(1)$$B\left(\omega \right)=\omega C_{d}-Y_{0}\cot\left(\beta h_{2}\right)$$
with
(2)$$Y_{0}={\left[\frac{120}{\sqrt{0.5\left(\varepsilon_{r}+1\right)}}\ln\left(1.079\frac{L}{R_{v}}\right)\right]}^{-1}$$

Here, *C_d_* represents the loaded capacitance between the semicircular patch and the top and bottom walls, and it can be approximated by $C_{d}\approx 0.5\varepsilon \pi R_{p}^{2}\left(1/h_{1}+1/h_{2}\right)$ with ε denoting the permittivity of the substrates. Note that since *h*_1_ is much smaller than *h*_2_ in our design, *h*_1_ predominates over *h*_2_ in determining the loaded capacitance. In addition, $\beta =\omega \sqrt{\varepsilon_{r}}/c_{0}$ is the phase constant with *c*_0_ standing for the speed of light in free space, and *Y*_0_ in (2) is the characteristic admittance of the short-circuited half coaxial-line model in Figure 6b. Upon enforcing *B*(*ω*) = 0 in (1) together with the approximation of cot(*βh*_2_) ≅ 1/*βh*_2_, the HMSICR’s resonance frequency, *f_r_*, can be approximated by
(3)$$f_{r}=\frac{1}{2\pi }{\left[\frac{Y_{0}c_{0}}{C_{d}\sqrt{\varepsilon_{r}}h_{2}}\right]}^{1/2}$$

In Figure 7, we provide the normalized resonance frequency (*f_r_*/*f*_0_) and the unloaded Q-factor (*Q_u_*) as functions of *R_p_* and *R_v_*, which are simulated using the commercial software package, HFSS [16]. For comparison, the *f_r_*/*f*_0_ curves obtained using (2) and (3) are also plotted in Figure 7a. The reasonable agreement between the curves simulated using HFSS and those calculated using (2) and (3) implies that (2) and (3) are useful empirical expressions for estimating the resonance frequency of the proposed HMSICR structure. In these figures, *R_v_* is fixed at 4 mm when *R_p_* is varying, and conversely, *R_p_* is fixed at 11 mm when *R_v_* is varying. Note that the HMSICR proposed here and the full-mode SICR presented in [13] are found to have a similar trend in the resonance-frequency variation, i.e., the larger the patch and the smaller the semicircular via-ring, the lower the resonance frequency. In addition, the HMSICR has a smaller unloaded *Q_u_* than the full-mode SICR, which is because the former has larger conduction losses in the semicircular patch and the shorting vias.

## 3. Sample Results

Shown in Figure 8a is the 3-D view of the proposed sample BPF consisting of four HMSICRs. The four HMSICRs are divided into two identical HMSICR sets. Resonators 1 and 2 are side-by-side juxtaposed to form a set, and resonators 3 and 4 form the other. Figure 8b–d, respectively, show the layouts of the bottom, middle, and top metal layers of an HMSICR set. The two HMSICR sets are then vertically stacked in a back-to-back configuration. The vertical stacking of the HMSICRs is carried out by flipping one HMSICR set upside down and placing it on the top of the other HMSICR set. The coupling between the two side-by-side HMSICRs depends on the window embedded in the cavities’ common sidewall and the etched H-shaped slot on the cavity’s bottom wall. On the other hand, the coupling between the two vertically stacked HMSICRs (i.e., between resonators 1 and 4, or between resonators 2 and 3) depends on the three fan-shaped apertures etched on their common top metal wall and the nearby semicircular patches. The input and output HMSICRs (i.e., resonators 1 and 4) are each fed by an SMA probe from the bottom to the top wall of the cavity.

The coupling between the two side-by-side HMSICRs (i.e., between resonators 1 and 2, or between resonators 4 and 3) is dominated by the magnetic coupling which is controlled by the H-shaped slot etched on the cavities’ bottom wall. It should be noted that this magnetic coupling mainly depends on the induced currents flowing on the semicircular patch’s shorting vias, and the dimension of the open window (the window having a width of 6.8 mm indicated in Figure 8c) has a minor effect on the coupling strength. Therefore, we adopt an H-shaped slot to restrain the magnetic flux for controlling the coupling strength. Figure 9 shows the coupling coefficient between resonators 1 and 2 as a function of *l* (the width of the middle section of the H-shaped slot indicated in the inset). A larger coupling coefficient denotes a stronger coupling strength. In the coupling-coefficient calculation, the slot width and the height of the H-shaped slot are fixed at 1.5 and 11.5 mm, respectively. The coupling between the top and bottom HMSICRs depends on the fan-shaped apertures on the common broadside wall and the nearby semicircular patches. These apertures simultaneously introduce both electrical and magnetic couplings, whose corresponding coupling coefficients are negative and positive, respectively [17,18,19,20,21,22]. Note that the total coupling is the sum of the electrical and magnetic couplings, and the net coupling coefficient may be positive or negative. The employed HMSICRs possess two natural resonance frequencies, 1.85 and 1.92 GHz, respectively, for the left and right HMSICRs shown in Figure 8. The coupling coefficients (denoted by *k_ij_* with the sub-indices *i, j* = 1, 2, 3, 4) are calculated to be *k*_12_ = *k*_34_ = 0.135, *k*_24_ = *k*_13_ = 0, *k*_23_ = 0.054, and *k*_14_ = –0.081, and the extracted external quality factor *Q_e_* is 14.3.

Figure 10 shows the measured and simulated frequency responses of the proposed BPF given in Figure 8. The zoom-in view around the passband region is also shown in this figure for clarity. Although compactly clustered, the four dips in the simulated S_11_ curve can still be distinguished. These four dips in the simulated S_11_ curve signify that the proposed BFP is a fourth-order design. However, from the measured S_11_ curve, we can hardly discriminate the four dips, and this might be due to the imprecise measurement resolution. This phenomenon can also be observed for the sample BPFs in [13]. The transmission zeros observed by the two passband edges are due to the well-known cross-coupling effect attributed to the circuit’s quadruplet topology [23,24,25,26]. The obtained measured (simulated) fractional BW (FBW) is 14.8% (14%) with the mid-band frequency of 1.81 (1.8) GHz. The measured (simulated) minimum in-band insertion loss (IL) is 0.6 (0.4) dB and the measured (simulated) TZs by the two passband edges are at the frequencies of 1.27 (1.22) and 2.28 (2.33) GHz.

The circuit performance in terms of the mid-band frequency (*f_c_*), FBW, IL, upper stopband BW (USB BW), and especially the circuit size is summarized in Table 1 to compare with recent size-reduced SIW (or SIW cavity) BPFs. In the comparison table, our proposed fourth-order BPF occupies an area of only 0.24 *λ**_d_* × 0.22 *λ**_d_* with a maximum in-band IL of 0.6 dB and a FBW of 14.8%. Obviously, our BPF has a better size efficiency than other circuits in the table. Note that the fourth-order BPF in Figure 6 of [13] in Table 1 has a circuit size of only 0.26 *λ**_d_* × 0.26 *λ**_d_*, and the corresponding size efficiency is closest to ours. In that BPF, the patch is shorted to the bottom wall using only one blind via instead of a circular array of blind vias. Although this strategy is recognized to very efficiently lower the resonance frequency of an SIW cavity, the BPF in Figure 6 of [13] suffers a relatively high IL. This is because the conduction loss associated with the patch’s current flowing to the bottom wall through only a single blind via is larger than that through an array of blind vias. Photos of the experimental circuit in this paper are given in Figure 11.

## 4. Conclusions

In this paper, we present the miniaturized design of the half-mode substrate integrated coaxial resonator BPF. For a fourth-order SIW-related BPF, the one proposed here has achieved an excellent circuit-area efficiency since it occupies a circuit area of only 0.24 *λ_d_*
*×* 0.22 *λ_d_* (physical dimensions of 27 *×* 24 mm^2^). The measured FBW is 14.8% with the minimum in-band IL of 0.6 dB. With such an excellent size-reduction achievement, the coaxial-mode-operated HMSICR BPF in this miniaturized design is believed very suitable for commercial wireless-communication applications.

## Figures and Tables

**Figure 1 micromachines-13-00389-f001:**
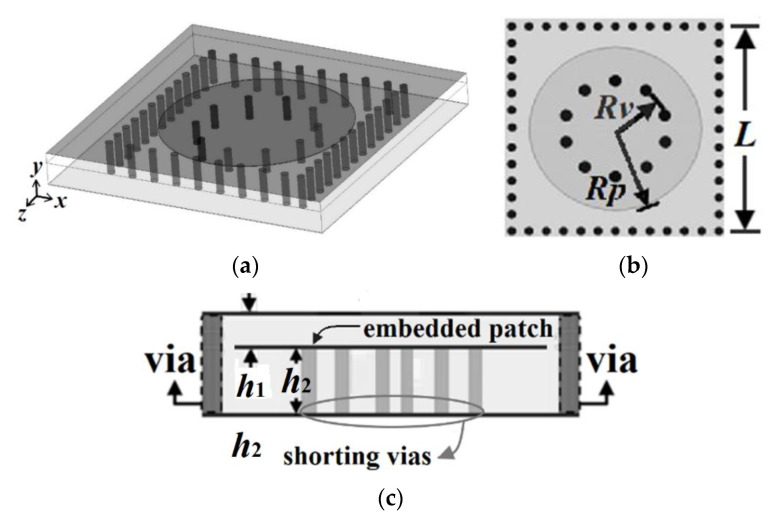
The full-mode SICR: (**a**) the 3-D view, (**b**) top transparent view, and (**c**) side view.

**Figure 2 micromachines-13-00389-f002:**
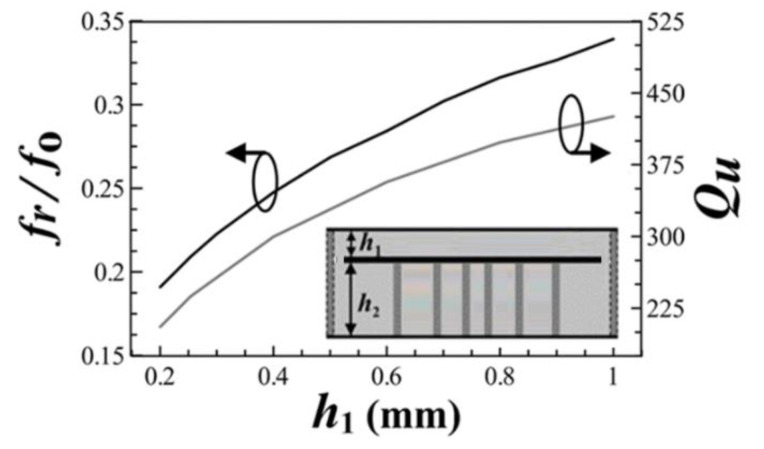
The simulated curves of normalized resonance frequency *f_r_*/*f*_0_ and *Q_u_* vs. the top substrate thickness *h*_1_ (with fixed *h*_2_ = 1.58 mm).

**Figure 3 micromachines-13-00389-f003:**
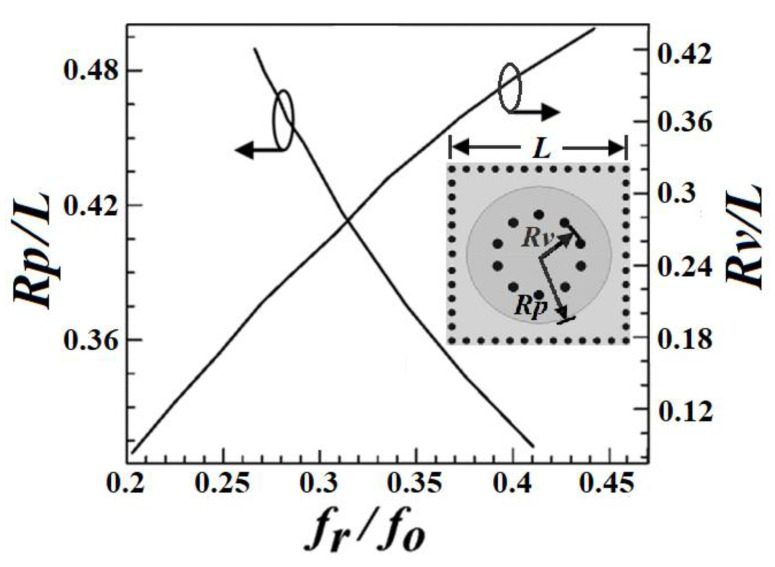
The simulated curves of the normalized resonance frequency *f_r_*/*f*_0_ vs. the dimension ratios of *R_p_*/*L* and *R_v_/L* (with fixed *L* = 24 mm).

**Figure 4 micromachines-13-00389-f004:**
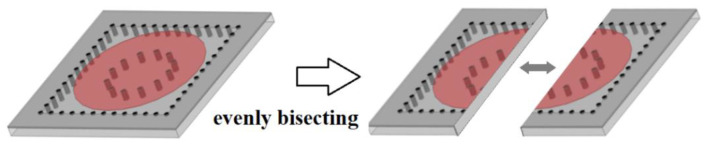
Evenly bisecting the full-mode SICR cavity to make the HMSICRs.

**Figure 5 micromachines-13-00389-f005:**
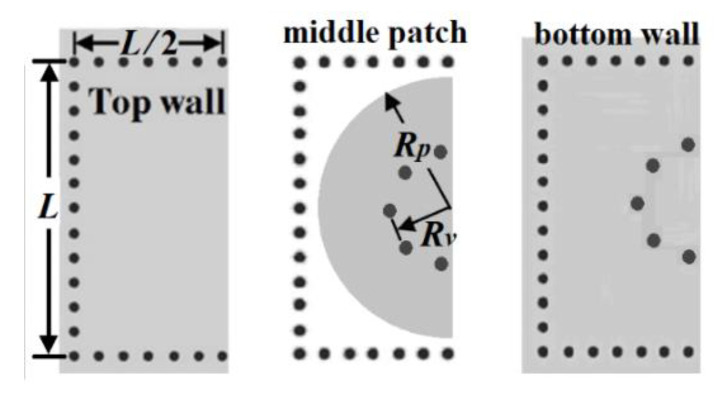
The layouts of the three metal layers for the proposed HMSICR cavity.

**Figure 6 micromachines-13-00389-f006:**
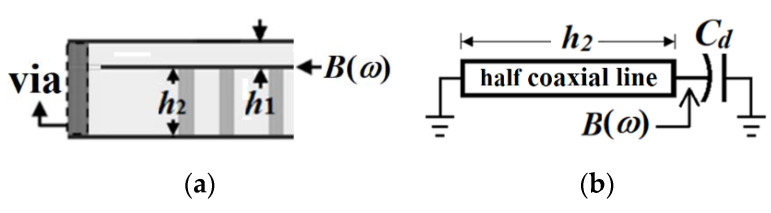
(**a**) The side view of the HMSICR; and (**b**) the equivalent transmission-line model for the susceptance looking into the embedded semicircular patch.

**Figure 7 micromachines-13-00389-f007:**
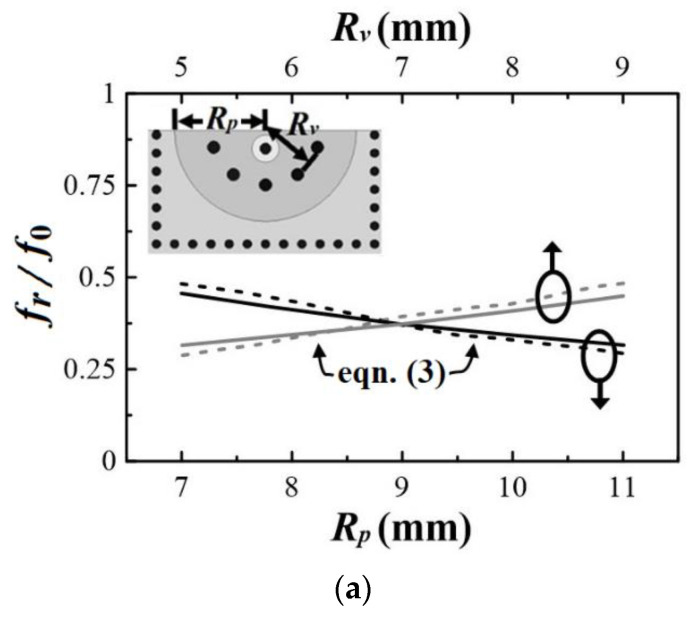
Simulated curves of (**a**) *f_r/_f*_0_ and (**b**) *Q_u_* vs. *R_p_* (radius of the embedded patch) and *R_v_* (radius of the semicircular via-ring).

**Figure 8 micromachines-13-00389-f008:**
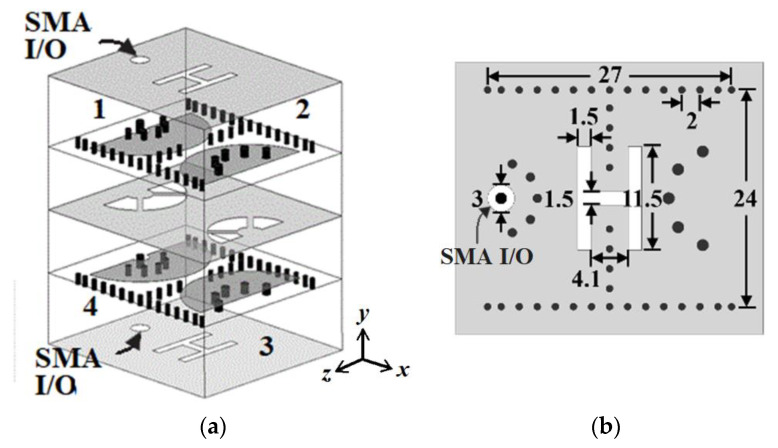
The proposed HMSICR BPF structure: (**a**) the 3-D view; layouts of the (**b**) bottom; and (**c**) middle, and (**d**) top metal layers.

**Figure 9 micromachines-13-00389-f009:**
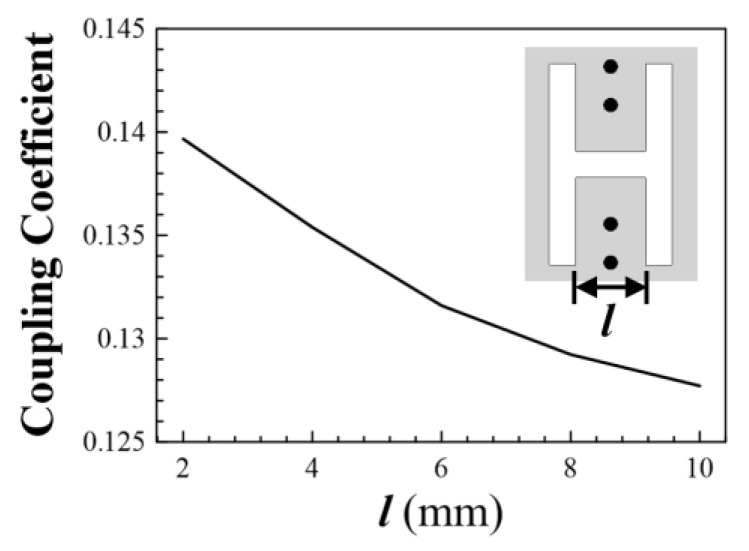
The coupling coefficient between resonators 1 and 2 as a function of the dimension *l* in the H-shaped slot.

**Figure 10 micromachines-13-00389-f010:**
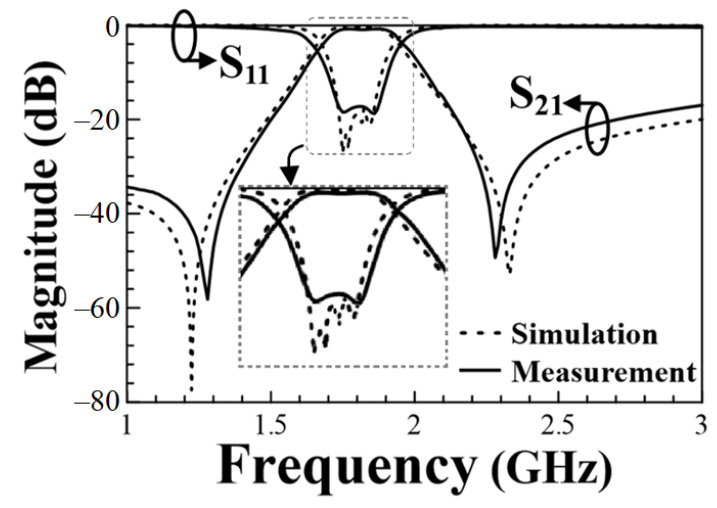
The measured and simulated frequency responses for the proposed sample HMSICR BPF.

**Figure 11 micromachines-13-00389-f011:**
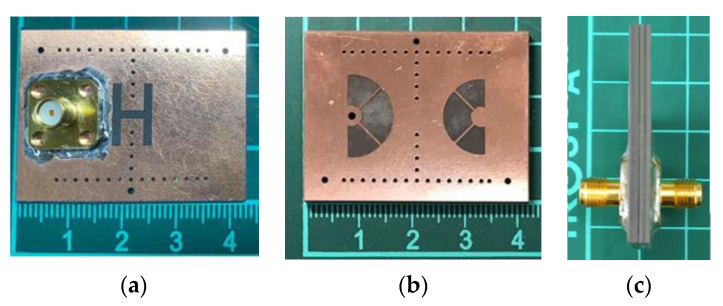
Photos of the experimental circuit: (**a**) the top view with one of the feeding SMAs; (**b**) the layer containing the coupling apertures; and (**c**) the side view.

**Table 1 micromachines-13-00389-t001:** The measured circuit performance comparison.

	f_c_ (GHz)	FBW (%)	IL (dB)	Circuit Order	Circuit Size (λ_d_ × λ_d_)	USB BW (S_21_ ≤ –20 dB)
[7] Figure 6	3.52	6.25	1.45	4	0.45 × 0.79	
[8] Figure 2b	8.79	40	1.2	5	1.15 × 0.33	
[10] Figure 12	5.2	38	0.74	3	0.99 × 0.99	
[12] Figure 6	3.58	6.5	1.7	3	0.35 × 0.47	~1.28 f_c_
[13] Figure 4	1.658	9.1	1.56	4	0.41 × 0.41	6.17 f_c_
[13] Figure 6	1.04	4.6	1.84	4	0.26 × 0.26	3 f_c_
This work	1.81	14.8	0.6	4	0.24 × 0.22	0.4 f_c_

*λ_d_*: the intrinsic wavelength (defined by $\lambda_{0}/\sqrt{\varepsilon_{r}}$ at frequency *f*_c_) in the dielectric medium.

## Data Availability

The data presented in this study are available on request from the corresponding author.

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
