# Peer review of "Miniaturized Band Pass Filter Design Using Half Mode Substrate Integrated Coaxial Resonators"

_micromachines, 2022, doi:10.3390/mi13030389_

Round 1

Reviewer 1 Report

The main idea of the proposed design of the Half-Mode Substrate Integrated Coaxial Resonator (HMSICR) is to make it more miniature compared to a conventional integrated coaxial resonator. It appears that, according to the results obtained, the goal has not been achieved:

  1. The author states that a 4-th order BPF is proposed, which provides better size efficiency in in circuit area. But the frequency characteristics for the proposed HMSICR BPF shown in Figure 7 correspond to 2-pole BPF with additional transmission zeros. Thus, this means that the structure of the HMSICR of the PPF Fig. 5 (c), consisting of two HMSICR, can be represented as a single resonator. Probably the same response could be achieved by using two coupled single cavity resonators. Here the author's comments are necessary to clarify the results obtained.
  2. I would like to recommend the authors to present the results of the modules of the reflection and transmission coefficients of the dual HSICRS compared with the result of a single integrated coaxial resonator.
  3. It would also be great to study the influence of some resonator parameters on frequency characteristics compared with the result of a single integrated coaxial resonator.
  4. In the table it is incorrect to compare the insertion losses of BPF having a different order.
  5. The same with the dimensions. If we compare the proposed filter design of 0.24 λd 0.22 λd with the original resonator structure [13], where four-pole PPFs were discussed, then the half-side has similar dimensions, but at the same time a higher order.
  6. Please specify at Figure 5. the input/output. Also, Figure 9.: what about the second SMA?

Author Response

We thank the reviewer for his effort in reviewing this manuscript. A rebuttal letter is uploaded to reply to the reviewer.

Reviewer 2 Report

1) Please provide more background about the systems you are designing and developing. The principal goal of a scientific publication is to educate the readers on your field and your discoveries. Start by describing the application you are targeting, and then proceed to explain how your work is improving the state-of-the-art.

2) Please improve the quality of Fig. 1, 2, and 3 – it is difficult to understand how you designed your system.

3) Fig. 4 is unreadable – it is not clear which curves are referred to fr/f0 and which ones are referred to Qu

4) Please re-arrange your paper such that if follows a reasonable flow: concept (with equations), simulations, fabrication, testing and validation. 

5) English language must be improved throughout the entire manuscript.

Author Response

We thank the reviewer for his effort in reviewing this manuscript. A rebuttal letter is uploaded to reply to the reviewer's comments.

Round 2

Reviewer 1 Report

The revised paper could be accepted for publication.

Reviewer 2 Report

No further comments